# Short- and Long-Term Chest-CT Findings after Recovery from COVID-19: A Systematic Review and Meta-Analysis

**DOI:** 10.3390/diagnostics14060621

**Published:** 2024-03-14

**Authors:** Mustufa Babar, Hasan Jamil, Neil Mehta, Ahmed Moutwakil, Tim Q. Duong

**Affiliations:** 1Department of Radiology, Albert Einstein College of Medicine and Montefiore Medical Center, Bronx, NY 10461, USA; mustufa.babar@einsteinmed.edu (M.B.); neil_mehta@brown.edu (N.M.); ahmedmoutwakil@alumni.stanford.edu (A.M.); 2Division of Surveillance and Policy Evaluation, National Cancer Center Institute for Cancer Control, Tokyo 104-0045, Japan; jhasan@ncc.go.jp; 3School of Public Health, St. Luke International University, Tokyo 104-0044, Japan

**Keywords:** SARS-CoV-2, long COVID, post-acute COVID-19 syndrome (PASC), computed tomography, pulmonary sequela

## Abstract

While ground-glass opacity, consolidation, and fibrosis in the lungs are some of the hallmarks of acute SAR-CoV-2 infection, it remains unclear whether these pulmonary radiological findings would resolve after acute symptoms have subsided. We conducted a systematic review and meta-analysis to evaluate chest computed tomography (CT) abnormalities stratified by COVID-19 disease severity and multiple timepoints post-infection. PubMed/MEDLINE was searched for relevant articles until 23 May 2023. Studies with COVID-19-recovered patients and follow-up chest CT at least 12 months post-infection were included. CT findings were evaluated at short-term (1–6 months) and long-term (12–24 months) follow-ups and by disease severity (severe and non-severe). A generalized linear mixed-effects model with random effects was used to estimate event rates for CT findings. A total of 2517 studies were identified, of which 43 met the inclusion (N = 8858 patients). Fibrotic-like changes had the highest event rate at short-term (0.44 [0.3–0.59]) and long-term (0.38 [0.23–0.56]) follow-ups. A meta-regression showed that over time the event rates decreased for any abnormality (β = −0.137, *p* = 0.002), ground-glass opacities (β = −0.169, *p* < 0.001), increased for honeycombing (β = 0.075, *p* = 0.03), and did not change for fibrotic-like changes, bronchiectasis, reticulation, and interlobular septal thickening (*p* > 0.05 for all). The severe subgroup had significantly higher rates of any abnormalities (*p* < 0.001), bronchiectasis (*p* = 0.02), fibrotic-like changes (*p* = 0.03), and reticulation (*p* < 0.001) at long-term follow-ups when compared to the non-severe subgroup. In conclusion, significant CT abnormalities remained up to 2 years post-COVID-19, especially in patients with severe disease. Long-lasting pulmonary abnormalities post-SARS-CoV-2 infection signal a future public health concern, necessitating extended monitoring, rehabilitation, survivor support, vaccination, and ongoing research for targeted therapies.

## 1. Introduction

The coronavirus disease 2019 (COVID-19) pandemic has been a watershed moment in global health, causing unprecedented strain on individual well-being and healthcare systems [1]. As the catastrophic waves of severe acute respiratory syndrome coronavirus 2 (SARS-CoV-2) infection begin to subside, the long-term consequences of the virus are coming into focus. There is mounting evidence that COVID-19 has effects, such as chronic cough, dyspnea, increased susceptibility to pulmonary infections, and intolerance to exercise, that persist well beyond the acute phase, commonly referred to as “long COVID” [2,3,4,5]. A recent meta-analysis revealed that about one-third of non-hospitalized patients, and more than half of hospitalized patients, reported persistent symptoms up to a year post-COVID [6].

Although the exact etiology of long COVID is currently unknown, it has been hypothesized to occur as a result of potential long-term tissue damage due to pulmonary-cardiovascular compromise, sepsis, and pathological inflammation during the acute and subacute phases of COVID-19 [7,8]. Lung damage, in particular, may play a significant role in the development of long COVID, since respiratory symptoms, such as cough, chest pain, and dyspnea are common presenting symptoms in those infected with SARS-CoV-2 [8,9,10,11]. Residual pulmonary damage could also contribute to the development to new clinical disorders (such as cardiac disorders, hypertension, diabetes, and renal disorders) as well as the worsening of pre-existing clinical disorders among individuals with COVID-19 compared to those of matched controls [12,13,14,15,16,17,18,19,20], which has broad health implications. Thoracic imaging, such a computed tomography (CT), can be used to evaluate the residual pulmonary effects from COVID-19 and provide valuable insights into the long-term morphological changes in the respiratory system, with ground-glass opacities (GGO), consolidations, and fibrosis being characteristic features frequently identified in the chest imaging of individuals with acute COVID-19 [8,21,22,23,24,25,26,27,28,29,30,31,32,33,34,35]. Emerging evidence suggests that the above-mentioned lingering respiratory symptoms are often accompanied by distinct CT findings, providing a visual narrative of the protracted aftermath of SARS-CoV-2 infection.

Numerous studies have reported CT lung abnormalities post-COVID [22,36,37], including a limited number of reviews and meta-analyses [8,38,39,40,41]. However, only a few studies covered multiple timepoints post-COVID [39,40] or were stratified by COVID-19 disease severity [40], and none covered beyond 12 months post-COVID. This meta-analysis of CT lung abnormalities post-COVID aims to build on prior meta-analyses by including more recent chest CT studies, studies at longer durations (up to 2 years post-COVID), as well as stratifying findings at multiple follow-ups and by COVID-19 disease severity. Our findings provide further insights into persistent lung abnormalities that could help inform clinical decision making and guide future research.

## 2. Materials and Methods

### 2.1. Protocol and Registration

The systematic review and meta-analysis was performed in accordance with the Preferred Reporting Items for Systematic Reviews and Meta-Analyses (PRISMA) statement [42]. The protocol was registered in the International Prospective Register of Systematic Reviews (PROSPERO: CRD42023447766). 

### 2.2. Eligibility Criteria

The inclusion criteria were as follows: (1) studies that included adult patients who recovered from acute COVID-19, confirmed by a SARS-CoV-2–positive reverse-transcription polymerase chain reaction test via nasopharyngeal swabs; (2) prospective or retrospective cohort studies, or cross-sectional studies; and (3) studies that included follow-up chest CT at least 12 months post-infection. Case reports, small case series (N < 10 patients), conference abstracts, and studies not in English were excluded.

### 2.3. Search Strategy

PubMed/MEDLINE was used to systematically search for relevant articles from 1 January 2020 to 23 May 2023. The search strategy included the following terms: ((“COVID” OR “COVID-19” OR “Coronavirus” OR “Coronavirus disease” OR “Coronavirus disease 2019” OR “SARS-CoV-2” OR “CoV-2” OR “SARS-CoV” OR “SARS” OR “Severe acute respiratory syndrome” OR “2019-nCoV” OR “nCoV” OR “Novel coronavirus”) AND (“Long-COVID” OR “Post-COVID” OR “Follow-up” OR “Long-term” OR “Chronic” OR “sequelae”) AND (“Computed tomography” OR “CT” OR “Chest CT”)) NOT (Review [Publication Type])).

After removing duplicates, two authors independently reviewed the search results using Covidence [43] and selected studies based on the inclusion criteria. Relevant studies were further identified through a manual search of secondary sources, including references of initially identified articles and reviews. After a full-text review, studies that met our eligibility criteria were included. Disagreements were resolved through consensus.

### 2.4. Data Extraction

Two authors independently extracted the data for study characteristics (author, year of publication, country, study design, percentage of patients with chest CT at long-term follow-up, longest follow-up time), patient characteristics (total sample size, age, sex, smoking habits, comorbidities), and chest-CT findings (any abnormalities, GGO, reticulation, consolidation, interlobular septal thickening, bronchiectasis, honeycombing, and fibrotic-like changes (combination of GGO, reticulation, bronchiectasis, and/or honeycombing)). Disagreements were resolved through consensus. 

### 2.5. Meta-Analysis

#### 2.5.1. Data Processing

Observational time intervals for CT findings were harmonized into monthly units. Time expressed in days was converted by a factor of 30, and when provided as a range, the midpoint was used for standardization. These intervals were aggregated into two broad temporal categories: short-term (≤6 months) and long-term (≥12 months). 

The severe group was reported for patients with “severe” or “critical” COVID-19 disease severity, and the non-severe group was reported for patients with “mild” or “moderate” COVID-19 disease severity. Individuals who had any of the various signs and symptoms of COVID-19 but did not have shortness of breath, dyspnea, or abnormal chest imaging were classified as having “mild” disease. Individuals who showed lower respiratory disease during clinical assessment or imaging and who had an oxygen saturation ≥ 94% on room air at sea level were classified as having “moderate” disease. Individuals who had an oxygen saturation < 94% on room air at sea level, a ratio of an arterial partial pressure of oxygen to fraction of inspired oxygen < 300 mm Hg, a respiratory rate > 30 breaths/min, or lung infiltrates > 50% were classified as having “severe” disease. Finally, individuals who had respiratory failure, septic shock, and/or multiple organ dysfunctions were classified as having “critical” disease [44]. Cases that were not clearly “non-severe” or “severe” were categorized as “mixed”.

#### 2.5.2. Statistical Analysis

A generalized linear mixed-effects model (GLMM) with a random-effects component was utilized to estimate pooled event rates for lung abnormalities. Logit transformation with a continuity correction of 0.5 for the zero event effect sizes was applied to individual study proportions to stabilize variances. Confidence intervals for individual studies were calculated using the Clopper–Pearson method. These estimations were made separately for the short- and long-term categories to avoid dependency between effect sizes. If a study reported multiple event rates in a time interval, only the last one was included in the GLMM model. Due to limited data points in some instances, the I^2^ statistic for heterogeneity was not always calculable. The data was represented in forest plots and figures after returning them to the original scale.

Meta-regression was utilized to inspect the impact of the time on the prevalence of lung abnormalities. For subgroup analysis, data classified as “mixed” for severity was excluded to focus on the “non-severe” and “severe” classifications. The *p*-value of Cochren’s Q was reported to indicate if there is a significant difference between the subgroups at the same time period. Statistical significance between CT findings at the 12- and 24-month follow-ups was calculated using a chi-square test. Meta-regression was also used in each severity strata to inspect relationships between the prevalence of lung abnormalities and time since a diagnosis of COVID-19. Statistical analysis was performed using the R statistical programming environment, version 4.3.1. Package meta, version 6.5, was used for all the meta-analysis elaborations. 

### 2.6. Quality Assessment

The quality of each included study was critically appraised by two authors using the validated risk of bias tool by Hoy et al. [45], which comprises 10 items and a summary assessment. Items 1 to 4 assess the external validity of the study (selection and nonresponse bias), and items 5 to 10 assess the internal validity (items 5 to 9 assess measurement bias, and item 10 assesses bias related to the analysis). The final score for each study was categorized into three classes: 0–3, 4–6, and 7–9, indicating low, moderate, and high risk of bias, respectively.

To evaluate the presence of publication bias, funnel plots were generated for each pooled event rate of lung abnormalities. Publication bias was visually assessed through funnel plots.

## 3. Results

### 3.1. Study Selection and Characteristics

A total of 2517 studies were identified, of which 43 met the inclusion (N = 8858 patients) (Figure 1) [46,47,48,49,50,51,52,53,54,55,56,57,58,59,60,61,62,63,64,65,66,67,68,69,70,71,72,73,74,75,76,77,78,79,80,81,82,83,84,85,86,87,88]. The majority of studies were from China (14 studies [51,58,59,60,61,64,65,66,74,83,85,86,87,88], 32.6%) or Italy (13 studies [46,47,48,49,50,52,54,55,56,70,75,77,84], 30.2%), and were prospective in nature (41 studies [46,47,48,49,50,51,53,54,56,57,58,59,60,61,62,63,64,65,66,67,68,69,70,71,72,73,74,75,76,77,78,79,80,81,82,83,84,85,86,87,88], 95.3%). 

Patients were infected with SARS-CoV-2 between December 2019 and December 2021. The median age of the patients was 60.3 years (57–63), with 61.5% being males and 37.8% being current/former smokers. Thirty-nine studies [47,49,50,51,52,53,54,55,56,57,58,59,60,61,62,63,65,66,67,68,70,71,72,73,74,75,76,77,78,79,80,81,82,83,84,85,86,87,88] reported outcomes stratified by disease severity; 17 for non-severe disease, 31 for severe disease, and 7 for mixed disease severity (Table 1). The most common comorbidities were hypertension (35.8%), cardiovascular disease, (28%), and obesity (24.9%) (Appendix A).

Of the 8858 patients included, 4223 (48%) had a chest CT at a short-term follow-up (median 3 months [3,4,5,6]) and 4872 patients (55%) had chest CT at long-term follow-up (median 12 months [12–12]). An overview of the lung abnormalities at long-term follow-ups is shown Table 2.

### 3.2. Pooled Event Rates of Follow-Up Chest-CT Lung Abnormalities over Time for Entire Population

The interval between 6 and 12 months was not included in the pooled effect-sizes’ synthesis analysis because of an inadequate number of effect sizes but was utilized in the meta-regression. Figure 2 illustrates the pooled event rates of each CT lung abnormality over time. The forest plots are reported in Appendix A and summarized in Appendix A. Meta-regression analysis using the number of months since the diagnosis of COVID-19 as the predictor variable are reported in Appendix A and summarized in Appendix A. Subgroup analysis comparing chest-CT findings between the 12- and 24-month follow-ups is reported in Appendix A.

#### 3.2.1. Short-Term Follow-Up (1 to 6 Months)

Thirty-two studies reported sizes in the short-term follow-up (median 3 months [3,4,5,6], range 1–6 months). CT findings at this follow-up revealed a high prevalence of lung abnormalities. The pooled event rate of any abnormality was 0.75 (0.63–0.84) with high heterogeneity (I^2^: 0.89). Fibrotic-like changes were the most common abnormality, with a pooled event rate of 0.44 (0.3–0.59) and high heterogeneity (I^2^: 0.9). GGO followed closely, with a pooled event rate of 0.43 (0.32–0.55) and also with high heterogeneity (I^2^: 0.94). Honeycombing had the lowest pooled event rate of 0.03 (0.02–0.07), and I^2^ could not be estimated due to an insufficient number of effect sizes.

#### 3.2.2. Long-Term Follow-Up (12 to 24 Months)

In 43 studies, long-term follow-up (median 12 months [12–12], range 12–24 months) showed a decrease in the pooled event rate of any abnormality to 0.63 (0.49–0.75) with high heterogeneity (I^2^: 0.95). GGO decreased to 0.25 (0.17–0.35) with high heterogeneity (I^2^: 0.93). Other abnormalities showed similar but smaller declining trends, except for honeycombing, which slightly increased to 0.04 (0.02–0.07) with low heterogeneity (I^2^: 0.4).

#### 3.2.3. Temporal Trends in Chest-CT Lung Abnormalities

In the meta-regression analysis, any abnormality and GGO significantly decreased over time (β = −0.137, *p* = 0.002 and β = −0.169, *p* < 0.001, respectively). In contrast, honeycombing was associated with an upward trend (β = 0.075, *p* = 0.03). The other lung abnormalities did not show significant associations (*p* > 0.05 for all).

A total of 37 studies [47,48,49,50,51,52,53,54,56,57,60,62,63,64,65,66,67,68,69,70,71,72,73,74,75,76,78,79,80,81,82,83,84,85,86,87,88] reported chest-CT findings at 12 months, while only 3 studies [55,59,61] reported findings at 24 months. When comparing CT abnormalities between the 12- and 24-month follow-ups, consolidation (12 months: 3.6% vs. 24 months: 0.9%, *p* = 0.036) and interlobular septal thickening (12 months: 17.3% vs. 24 months: 7%, *p* = 0.043) significantly decreased over time. The other lung abnormalities showed no significant change between the 12- and 24-month follow-ups.

### 3.3. Pooled Event Rates of Follow-Up Chest-CT Lung Abnormalities over Time with COVID-19 Severity as the Mediator

The severity subgroup estimates for each chest-CT finding for the short- and long-term follow-ups are illustrated in Figure 3, and the forest plots for the conducted meta-analysis are shown in Appendix A and summarized in Appendix A. Meta-regression analysis for each severity strata using the number of months since the diagnosis of COVID-19 as the predictor variable is reported in Appendix A and summarized in Appendix A.

#### 3.3.1. Non-Severe Subgroup

In the non-severe subgroup, event rates for various abnormalities (any abnormality, GGO, consolidation, and interlobular septal thickening) decreased from the short-term to the long-term follow-up and was supported by negative trends in meta-regression (*p* < 0.05 for all). Additionally, the event rates for fibrotic-like changes, bronchiectasis, and reticulation remained stable or slightly decreased over time, but meta-regression did not show a significant trend (*p* > 0.05 for all). Honeycombing was negligible at both timepoints, and meta-regression could not be calculated due to insufficient data. At the short-term follow-up, I^2^ ranged between 0.53 and 0.90, with interlobular septal thickening having the lowest I^2^ and GGO having the highest I^2^. At the long-term follow-up, I^2^ ranged between 0.64 and 0.93, with consolidation having the lowest I^2^ and fibrotic-like changes having the highest I^2^.

#### 3.3.2. Severe Subgroup

In the severe subgroup, event rates for any abnormality, GGO, and consolidation decreased from the short-term to the long-term follow-up and was supported by negative trends in meta-regression (*p* < 0.05 for all). Additionally, the event rates for fibrotic-like changes, bronchiectasis, reticulation, interlobular septal thickening, and honeycombing remained stable or slightly decreased over time, but meta-regression did not show a significant trend (*p* > 0.05 for all). At the short-term follow-up, I^2^ ranged between 0.83 and 0.96, with fibrotic-like changes having the lowest I^2^ and bronchiectasis having the highest I^2^. At the long-term follow-up, I^2^ ranged between 0.60 and 0.96, with honeycombing having the lowest I^2^ and bronchiectasis having the highest I^2^.

#### 3.3.3. Comparison between Severity Subgroups

The severe subgroup had significantly higher event rates for any abnormalities (*p* = 0.01), bronchiectasis (*p* < 0.001), and reticulation (*p* = 0.01) at the short-term follow-up, and any abnormalities (*p* < 0.001), bronchiectasis (*p* = 0.02), fibrotic-like changes (*p* = 0.03), and reticulation (*p* < 0.001) at the long-term follow-up when compared to the non-severe subgroup.

### 3.4. Quality Assessment 

The risk-of-bias assessment is shown in Appendix A. Thirteen studies (30.2%) had a moderate risk of bias while the remaining 30 studies (69.8%) had a low risk of bias. The items that had the highest scores were those that were tested for external validity (items 1 to 5), with item 2 and item 3 having the highest summative scores of 39/43 and 42/43, respectively. A visual inspection of the funnel plots revealed publication bias in some CT findings (Appendix A).

## 4. Discussion

Our meta-analysis of 43 studies, stratified by COVID-19 severity, revealed significant CT abnormalities up to 2 years after SARS-CoV-2 infection. While some abnormalities like GGO and consolidation decreased over time, others including fibrotic-like changes, bronchiectasis, reticulation, and interlobular septal thickening remained unchanged. Notably, honeycombing increased over time. Patients with severe COVID-19 exhibited higher incidences of any abnormality, bronchiectasis, fibrotic-like changes, and reticulation up to 2 years post-COVID compared to those with non-severe COVID-19.

At the 12- and 24-month follow-ups, the only chest-CT abnormalities that showed significant improvement were consolidation and interlobular septal thickening. This underscores the persistent nature of lung abnormalities over years following the initial infection. However, the results should be interpreted with caution because of the small number of studies.

Although there have been a few prior meta-analyses of CT abnormalities in post-COVID-19 patients [38,39,40,41], our meta-analysis is the largest to date (43 studies) and is stratified by COVID-19 disease severity and multiple timepoints up to 2 years post-COVID. There is only one other meta-analysis that investigated chest CT up to 1 year post-COVID-19. Watanabe et al. [40] included 15 studies and found that residual CT abnormalities were common in hospitalized COVID-19 patients 1 year after recovery, especially for fibrotic-like changes in those with severe/critical severity. We also observed fibrotic-like changes, along with bronchiectasis and reticulation, to be greater in those with severe/critical severity. Watanabe et al. [40] reported a 21% prevalence of any abnormality at long-term follow-up for mild/moderate disease and 38% for severe/critical disease. However, in our study, we found approximately double these rates: 39% for mild/moderate disease and 75% severe/critical cases. These discrepancies may be due to differences in patient demographics (age, sex, race, comorbidities) when infection occurred, the duration of the follow-up, and the CT-in-utilization rate. In particular, our cohort had a higher median age (60.3 vs. 56 years old), percentage of males (61.5% vs. 51.3%), proportion of cardiovascular disease (28% vs. 7.5%), and longer duration of patient follow-up (up to 2 years vs. up to 1 year).

Post-COVID pulmonary fibrosis, with an incidence ranging between 5 and 75%, contributes to the burden of chronic respiratory issues among survivors [89]. Its pathophysiology likely stems from the local proinflammatory environment caused by macrophage and immune cell infiltration in the lungs, which disrupts the natural homeostatic tissue-repair functions [90]. Given that pulmonary fibrosis may represent permanent lung damage, identifying its risk factors is crucial for potential prophylactic interventions. Our study identified severe COVID-19 as a risk factor for residual pulmonary fibrosis, which is consistent with other studies [91,92]. Although there is no consensus on treatment, antifibrotic agents may benefit these patients [89].

The functional consequences of lingering CT abnormalities from long COVID remain uncertain [59]. Since these abnormalities suggest lung damage, they can potentially lead to chronic fatigue, post-exertional malaise, and a reduced quality of life [93,94]. Furthermore, the high prevalence of residual CT abnormalities in our study, raises concerns about the increased risk of new-onset pulmonary diseases, such as chronic obstructive pulmonary disease (COPD), asthma, pneumonia, and bronchitis, and compromised pulmonary–cardiovascular health, resulting in decreased exercise tolerance and increased fatigue. In addition, since there is a known association between immunology and cancer, pathological inflammation and immunological responses may increase the rate of lung cancer. Additionally, these radiological abnormalities may decrease the sensitivity of detecting lung cancer by obscuring certain details in imaging. These persistent lung abnormalities can also exacerbate pre-existing pulmonary diseases in the years to come [95]. Large population-based studies involving pulmonary-function testing and long-term follow-ups of at-risk patients with abnormal CT abnormalities are warranted [21,96].

An identification of the potential risk factors of long COVID is necessary to better understand who is at risk and to allow for early clinical support. A recent meta-analysis found certain demographics (female sex, older age, higher BMI, and smoking) and comorbidities (anxiety and/or depression, asthma, COPD, diabetes, ischemic heart disease, and immunosuppression) to be associated with an increased risk of long COVID, whereas vaccination had a protective role [97]. Besides taking preventative measures of receiving vaccination, individuals with risk factors and previous COVID-19 infection may require follow-up outpatient services to manage long COVID and explore the possible association between their symptoms and residual lung damage.

Similar to SARS-CoV-2, patients with SARS-CoV-1, Middle East respiratory syndrome (MERS), and influenza A exhibited residual pulmonary abnormalities at long-term follow-ups, especially fibrotic-like changes (SARS-CoV-2: 38% vs. SARS-CoV-1: 62% vs. MERS: 33% vs. influenza A: 42%) [98,99,100,101,102,103]. Genetic homology between SARS-CoV-2, SARS-CoV-1, and MERS suggests a genetically influenced fibroproliferative process contributing to increased risks of post-COVID pulmonary complications [104,105,106,107]. However, unlike other viruses, SARS-CoV-2 leads to a higher burden of extrapulmonary organ involvement, resulting in a higher level of health impairment during both the acute and post-acute phases [108]. Regarding symptoms, shortness of breath was less common in SARS-CoV-2 patients (17%) in comparison with SARS-CoV-1 (32%), MERS (51%), and influenza A (34%) [109,110]. Although the prevalence of pulmonary abnormalities (e.g., fibrotic-like changes) and symptoms may seem contradictory between the different viruses, these findings may be biased by reporting only confirmed cases and should therefore be considered when interpreting the data. However, unlike SARS-CoV-1 and MERS, which had a total of <11,000 combined confirmed cases, SARS-CoV-2 has an alarming of >700,000,000 confirmed cases to date [111,112]. The magnitude of cases with a persistence of pulmonary abnormalities is a matter of concern. Despite its high prevalence, SARS-CoV-2 has a significantly lower mortality rate (SARS-CoV-2: 2.1% vs. SARS-CoV-1: 9.5% vs. MERS: 34.4%) [113].

This study has several limitations. High heterogeneity in most of our pooled event-rate estimates suggest the presence of unaccounted mediating factors. Additionally, some studies did not report effect sizes for specific lung abnormalities, limiting our comprehensive analysis. Furthermore, limited observations in some severity groups at various time intervals hampered our precise assessment of initial infection severity on the evolution of specific abnormalities. Additionally, the CT abnormalities could not be graded according to their severity because there were only a few studies that graded the severity of the abnormalities, and the scoring system was neither standardized nor validated across studies. Therefore, it remains unclear how the abnormalities can be compared to those observed in a healthy cohort. Another limitation is the use of broad time brackets (1–6 months, 12–24 months) due to the limited number of suitable studies. Moreover, we did not collect pulmonary-function tests as this was not the focus of the study. Consequently, we were unable to correlate pulmonary function with imaging findings. Furthermore, a small percentage of patients had pre-existing pulmonary diseases, which may be a confounder. Finally, it is possible that some of the patients might have had pre-existing CT abnormalities prior to COVID-19.

## 5. Conclusions

Significant pulmonary CT abnormalities remained for up to 2 years post-COVID, especially in patients with severe disease. The sheer number of individuals infected with SARS-CoV-2 world-wide suggests that pulmonary sequela and related complications could be a major public-health issue in years to come. Our findings underscore the need for extended monitoring, rehabilitation, and support for COVID-19 survivors, vaccination for severe disease prevention, and ongoing research into targeted therapies to mitigate the enduring pulmonary consequences of SARS-CoV-2.

## Figures and Tables

**Figure 1 diagnostics-14-00621-f001:**
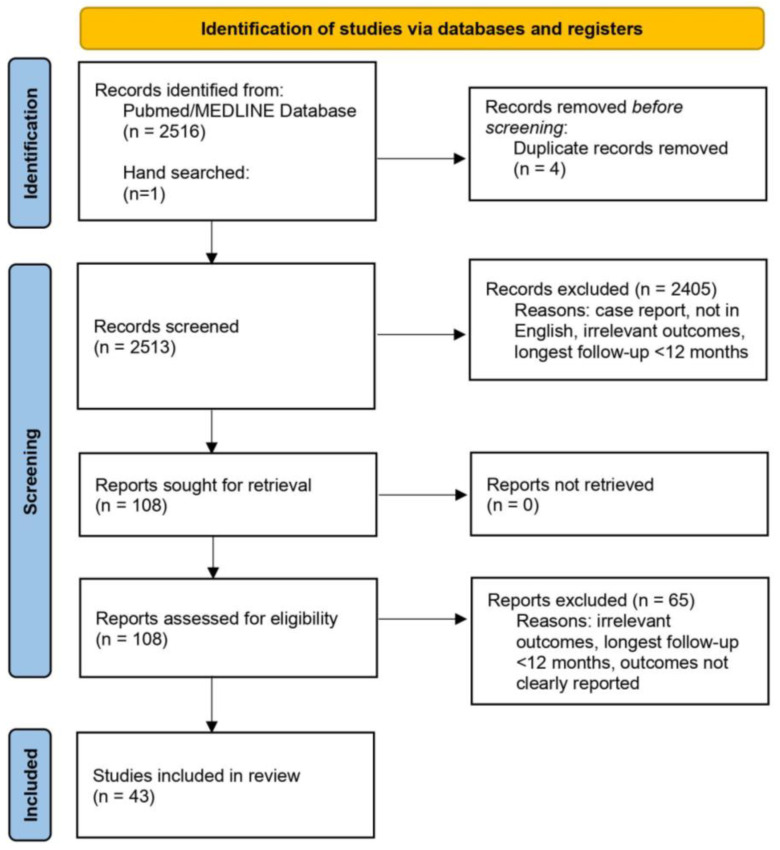
Flowchart of study selection. A total of 2513 records were screened, 108 were assessed for eligibility, and 43 were included in the analysis.

**Figure 2 diagnostics-14-00621-f002:**
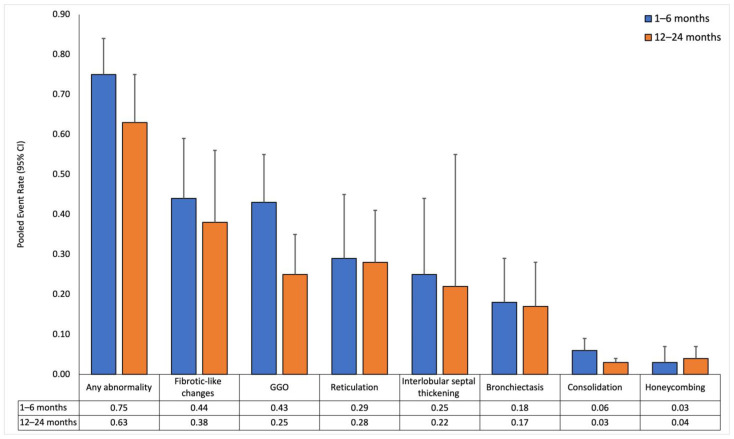
Pooled event rates of chest-CT findings over time for all patients. Data is reported at short-term (1–6 months) and long-term (12–24 months) follow-ups.

**Figure 3 diagnostics-14-00621-f003:**
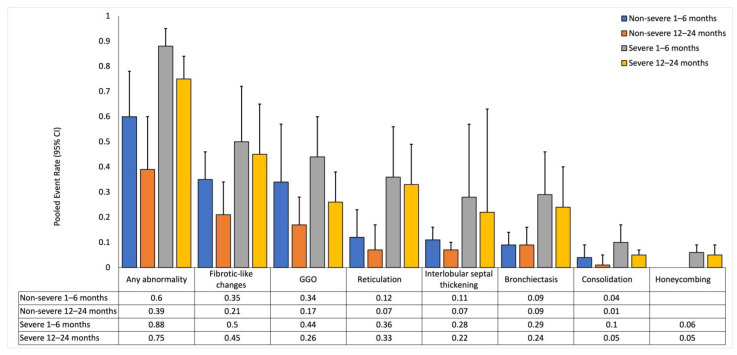
Pooled event rates for chest-CT findings with COVID-19 severity as the mediator. Data is reported for non-severe and severe subgroups at short-term (1–6 months) and long-term (12–24 months) follow-ups. Event rates for each severity subgroup is compared with that in the other severity subgroup at the respective follow-up. UE: Unable to estimate.

**Table 1 diagnostics-14-00621-t001:** Characteristics of included studies (n = 43). R: retrospective, P: prospective, H: hospitalized, NH: non-hospitalized, NR: not reported.

Author (Year)	Country	Study Design	Patients with Chest CT at Follow-Up, n (%)	Longest Follow-Up Time, Months	Initial-Infection Time Period		Patient Characteristics					
					Start Date	End Date	N	Male Sex, %	Age, Years	Ever Smoker, %	Disease-Severity Group(s)	Hospitalization Status
Bellan et al. (2021) [47]	Italy	P	190 (95)	12	1 March 2020	29 June 2020	200	61	Median 62 (51–71)	44.5	Severe	H
Zhan et al. (2021) [85]	China	P	121 (100)	12	15 January 2020	31 March 2020	121	41.3	Median 49 (40–57)	NR	Non-severe, severe	H
Zhou et al. (2021) [87]	China	P	97 (80.8)	12	29 January 2020	1 April 2020	120	40.8	Mean 51.6 (10.8)	13.3	Non-severe, severe	H
Li et al. (2021) [64]	China	P	141 (100)	12	28 December 2019	30 April 2020	141	63.1	Median 59.0 (51–66)	11.3	NR	H
Chen et al. (2021) [51]	China	P	36 (87.8)	12	1 February 2020	15 March 2020	41	58.5	Median 51 (38–59)	9.8	Mild, severe	H
Zhao et al. (2021) [87]	China	P	94 (100)	12	16 January 2020	6 February 2020	94	57.5	Mean 48.1 (11.9)	7.5	Mild, moderate, severe, critical	H
Gamberini et al. (2021) [56]	Italy	P	37 (20.8)	12	22 February 2020	4 May 2020	178	72.5	Median 64 (55–70)	NR	Severe	H
Han et al. (2021) [60]	China	P	62 (100)	12	NR	1 June 2020	62	54.8	Mean 57 (10)	NR	Severe	H
Wu et al. (2021) [83]	China	P	83 (100)	12	1 February 2020	31 March 2020	83	57.8	Median 60 (52–66)	0	Severe	H
Zangrillo et al. (2021) [84]	Italy	P	36 (64.3)	12	25 February 2020	27 April 2020	56	89.3	Mean 56 (11.9)	35.4	Severe	H
Faverio et al. (2022) [54]	Italy	P	270 (94.1)	12	1 March 2020	1 June 2020	287	74.2	Median 60.7 (53.4–68.8)	26.5	Severe	H
Rigoni et al. (2022) [75]	Italy	P	47 (10)	12	1 March 2020	1 May 2020	471	63.8	Median 71 (58–81)	NR	Mixed (mild/moderate/severe)	H
Liao et al. (2022) [65]	China	P	256 (84.5)	12	18 March 2021	30 April 2021	303	19.5	Median 39, (33–48)	3.3	Mild, moderate, severe, critical	H
González et al. (2022) [57]	Spain	P	41 (22.7)	12	1 March 2020	1 August 2020	181	66.9	Median 61 (52–67)	38.1	Critical	H
Corsi et al. (2022) [52]	Italy	P	63 (88.7)	12	25 February 2020	2 May 2020	71	36.7	Median 66 (59–73)	54	Severe	H
Zhang et al. (2022) [86]	China	P	204 (80)	12	1 January 2020	1 April 2020	255	51	Mean 43.8 (16.1)	13.7	Mild, moderate, severe, critical	H
Eberst et al. (2022) [53]	France	P	64 (75.3)	12	1 April 2020	1 June 2021	85	78.8	Median 68.4 (60.1–72.9)	58.8	Severe	H
Lorent et al. (2022) [67]	Belgium	P	105 (35.1)	12	1 March 2020	31 May 2020	299	68.6	Median 59 (52–68)	NR	Moderate, severe	H
Liu et al. (2022) [66]	China	P	486 (81.8)	12	10 February 2020	30 April 2020	594	46.3	Median 63 (53–68)	13	Moderate, severe, critical	H
Marando et al. (2022) [69]	Switzerland	P	31 (79.5)	12	1 March 2020	15 April 2020	39	79.5	Median 64.5 (52.7–72.2)	38.7	NR	H
Luger et al. (2022) [68]	Austria	P	91 (100)	12	29 April 2020	12 August 2020	91	61.5	Median 57 (51–70)	34	Mixed (mild/moderate/severe/critical)	H and NH
Pan et al. (2022) [74]	China	P	209 (100)	12	27 January 2020	31 March 2020	209	44.5	Mean 49 (13)	1.9	Severe, critical	H
Tarraso et al. (2022) [79]	Spain	P	156 (54.9)	12	1 May 2020	31 July 2020	284	55.3	Mean 60.5 (11.9)	42.3	Mild, moderate, severe	H
Vijayakumar et al. (2022) [82]	England	P	32 (100)	12	1 March 2020	1 June 2020	32	65.6	Mean 62 (11)	59	Mixed (mild/moderate/severe)	H
Martino et al. (2022) [70]	Italy	P	47 (73.4)	12	25 March 2020	15 May 2020	64	64.1	Median 68 (56.5–75)	43.6	Severe	H
Bocchino et al. (2022) [49]	Italy	P	84 (100)	12	1 March 2020	1 July 2021	84	66.7	Mean 61 (11)	42	Moderate	H
Huang et al. (2022) [61]	China	P	57 (4.8)	24	7 January 2020	29 May 2020	1192	54	Median 57.0 (48.0–65.0)	17	Moderate, severe, critical	H
Barini et al. (2022) [46]	Italy	P	115 (100)	18	1 March 2020	1 May 2020	115	67.8	Mean 60 (15)	NR	NR	H
van Raaij et al. (2022) [81]	Netherlands	P	66 (100)	12	23 March 2020	23 June 2020	66	69.7	Median 60.5 (54.0−69.3)	43.9	Moderate, severe	H
Lenoir et al. (2022) [62]	Switzerland	P	25 (4.3)	12	1 May 2020	31 December 2021	584	56.8	Mean 58.0 (14.1)	45	Mixed (non-severe/severe)	NR
Guo et al. (2022) [58]	China	P	95 (45.7)	18.5	NR	17 February 2020	208	48.1	Median 58 (50.0–64.3)	12	Mild, severe	H
Bernardinello et al. (2023) [48]	Italy	P	347 (100)	12	1 February 2020	1 April 2021	347	62.5	Median 63 (53–72)	37.8	NR	H
Han et al. (2023) [59]	China	P	144 (100)	24	20 January 2020	10 March 2020	144	55	Median 60 (27–80)	17	Mixed (moderate/severe/critical)	H
Bongiovanni et al. (2023) [50]	Italy	P	233 (100)	12	1 March 2020	1 April 2021	233	61.4	NR	42.1	Moderate, severe, critical	H
Lerum et al. (2023) [63]	Norway	P	124 (47.3)	12	NR	1 June 2020	262	58	Mean 58.6 (14.2)	41.9	Mild, moderate, severe	H
Sanna et al. (2023) [77]	Italy	P	19 (19)	15	1 March 2020	1 August 2020	100	62	Mean 59.6 (12.8)	39	Mixed (moderate/severe/critical)	H
Núñez-Fernández et al. (2023) [73]	Spain	P	70 (36.1)	12	NR	NR	194	55.8	Median 62 (51.5–71)	40.2	Severe	H
Mulet et al. (2023) [71]	Spain	P	126 (93.3)	12	NR	NR	135	61.5	Mean 61 (19)	37.8	Mixed (mild/moderate/severe)	H
Noureddine et al. (2023) [72]	France	P	60 (100)	12	1 April 2020	1 June 2020	60	78	Mean 64.6 (9.6)	56.7	Severe	H
Sahanic et al. (2023) [76]	Austria	P	101 (93.5)	12	1 April 2020	1 June 2020	108	64	Median 56 (49–68)	32	Mild, moderate, severe	H and NH
van der Sar-van der Brugge et al. (2023) [80]	Netherlands	P	66 (40.7)	12	1 March 2020	1 April 2020	162	59	Mean 65.5 (0.95)	54	Moderate, severe, critical	H
Schlemmer et al. (2023) [78]	France	P	123 (25.4)	12	10 March 2020	25 November 2020	485	73	Median 60.7 (53.4–67.6)	37.3	Severe, critical	H
Flor et al. (2023) [55]	Italy	P	18 (100)	24	1 February 2020	31 May 2020	18	83	Median 70 (65–78)	NR	Severe	H

**Table 2 diagnostics-14-00621-t002:** Chest-CT evaluation of residual lung abnormalities at long-term follow-up after COVID-19. NR: not reported.

Author (Year)	Longest Follow-Up Time, Months	Chest-CT Findings, n/N (%)							
		Any Abnormality	GGO	Fibrotic-like Changes	Reticulation	Consolidation	Interlobular Septal Thickening	Bronchiectasis	Honeycombing
Bellan et al. (2021) [47]	12	44/190 (23.1)	NR	NR	NR	NR	NR	NR	NR
Zhan et al. (2021) [85]	12	10/121 (8.3)	NR	NR	NR	NR	NR	NR	NR
Zhou et al. (2021) [87]	12	55/97 (56.7)	16/97 (16.5)	17/97 (17.5)	NR	NR	NR	14/97 (14.4)	NR
Li et al. (2021) [64]	12	13/25 (52)	6/25 (24)	NR	7/25 (28)	0/25 (0)	9/25 (36)	NR	NR
Chen et al. (2021) [51]	12	17/36 (47.2)	NR	NR	NR	NR	NR	NR	NR
Zhao et al. (2021) [87]	12	67/94 (71.3)	38/94 (40.4)	8/94 (8.5)	4/94 (4.3)	2/94 (2.1)	10/94 (10.7)	NR	NR
Gamberini et al. (2021) [56]	12	NR	21/37 (56.8)	26/37 (70.3)	13/37 (35.1)	3/37 (8.1)	NR	10/37 (27)	3/37 (8.1)
Han et al. (2021) [60]	12	45/62 (72.6)	7/62 (11.3)	35/62 (56.5)	32/62 (51.6)	6/62 (9.7)	28/62 (45.2)	27/62 (43.5)	NR
Wu et al. (2021) [83]	12	20/83 (24.1)	19/83 (22.9)	NR	3/83 (3.6)	NR	4/83 (4.8)	1/83 (1.2)	NR
Zangrillo et al. (2021) [84]	12	NR	NR	4/36 (11.1)	NR	NR	NR	NR	NR
Faverio et al. (2022) [54]	12	178/270 (65.9)	61/270 (22.6)	NR	98/270 (36.3)	8/270 (3)	NR	14/270 (5.2)	3/270 (1.1)
Rigoni et al. (2022) [75]	12	NR	23/47 (48.9)	NR	NR	1/47 (2.1)	45/47 (95.7)	13/47 (27.7)	NR
Liao et al. (2022) [65]	12	96/256 (37.5)	63/256 (24.6)	26/256 (10.2)	2/256 (0.8)	8/256 (3.1)	NR	4/256 (1.6)	NR
González et al. (2022) [57]	12	41/41 (100)	27/41 (65.9)	15/41 (36.6)	22/41 (53.7)	3/41 (7.3)	41/41 (100)	37/41 (90.2)	NR
Corsi et al. (2022) [52]	12	48/63 (76.2)	2/63 (3.2)	NR	38/63 (60.3)	2/63 (3.2)	NR	42/63 (66.7)	NR
Zhang et al. (2022) [86]	12	137/245 (55.9)	11/204 (5.4)	45/245 (18.4)	NR	1/245 (0.4)	13/245 (5.3)	NR	NR
Eberst et al. (2022) [53]	12	60/64 (93.8)	32/64 (53.3)	NR	51/64 (85)	NR	NR	44/64 (73.3)	3/64 (5)
Lorent et al. (2022) [67]	12	68/105 (64.8)	39/105 (37.1)	NR	58/105 (55.2)	1/105 (1)	NR	21/105 (20)	NR
Liu et al. (2022) [66]	12	NR	0/486 (0)	249/486 (51.2)	NR	NR	NR	22/486 (4.5)	NR
Marando et al. (2022) [69]	12	30/31 (96.8)	21/31 (67.7)	23/31 (74.2)	NR	3/31 (9.7)	NR	NR	NR
Luger et al. (2022) [68]	12	49/91 (53.8)	40/91 (44)	NR	39/91 (42.9)	1/91 (1.1)	NR	8/91 (8.8)	NR
Pan et al. (2022) [74]	12	53/209 (25)	50/209 (23.9)	NR	28/209 (13.4)	3/209 (1.4)	NR	14/209 (11.5)	NR
Tarraso et al. (2022) [79]	12	123/156 (78.8)	71/156 (45.5)	102/156 (65.4)	53/156 (33.9)	25/156 (16)	NR	48/156 (30.8)	NR
Vijayakumar et al. (2022) [82]	12	27/32 (84.4)	NR	NR	NR	NR	NR	NR	NR
Martino et al. (2022) [70]	12	30/47 (63.8)	7/47 (14.9)	7/47 (14.9)	19/47 (40.4)	7/47 (14.9)	5/47 (10.6)	4/47 (8.5)	2/47 (4.2)
Bocchino et al. (2022) [49]	12	6/84 (7.1)	2/84 (2.4)	4/84 (4.8)	2/84 (2.4)	0/84 (0)	NR	2/84 (2.4)	0/84 (0)
Huang et al. (2022) [61]	24	47/57 (82.5)	34/57 (59.6)	NR	1/57 (1.8)	2/57 (3.5)	4/57 (7)	NR	NR
Barini et al. (2022) [46]	18	NR	NR	NR	NR	NR	NR	17/115 (14.8)	NR
van Raaij et al. (2022) [81]	12	34/66 (51.5)	19/66 (28.8)	NR	14/66 (21.2)	3/66 (4.5)	NR	23/66 (34.8)	NR
Lenoir et al. (2022) [62]	12	NR	24/25 (96)	NR	11/25 (44)	3/25 (12)	NR	8/25 (32)	NR
Guo et al. (2022) [58]	18.5	NR	28/95 (29.5)	NR	34/95 (35.8)	NR	NR	NR	NR
Bernardinello et al. (2023) [48]	12	24/347 (6.9)	19/347 (5.5)	NR	NR	2/347 (0.6)	21/347 (6.1)	7/347 (2)	NR
Han et al. (2023) [59]	24	56/144 (38.9)	6/144 (4.2)	33/144 (22.9)	50/144 (34.7)	0/144 (0)	NR	23/144 (16)	8/144 (6)
Bongiovanni et al. (2023) [50]	12	140/233 (60.1)	39/233 (16.7)	74/233 (31.8)	NR	NR	NR	41/233 (17.6)	NR
Lerum et al. (2023) [63]	12	NR	62/124 (50)	74/124 (59.7)	37/124 (29.8)	8/124 (6.5)	17/124 (13.7)	NR	NR
Sanna et al. (2023) [77]	15	19/19 (100)	7/19 (36.8)	19/19 (100)	NR	0/19 (0)	0/19 (0)	0/19 (0)	0/19 (0)
Núñez-Fernández et al. (2023) [73]	12	NR	13/70 (18.6)	NR	21/70 (30)	NR	NR	20/70 (28.6)	NR
Mulet et al. (2023) [71]	12	46/125 (36.8)	31/125 (24.6)	37/125 (29.4)	NR	NR	NR	NR	NR
Noureddine et al. (2023) [72]	12	50/60 (83.3)	29/60 (48.3)	NR	42/60 (70)	NR	NR	35/60 (58.3)	3/60 (5)
Sahanic et al. (2023) [76]	12	52/101 (51.5)	NR	NR	NR	NR	NR	NR	NR
van der Sar-van der Brugge et al. (2023) [80]	12	33/66 (50)	31/66 (47)	16/66 (24.2)	NR	NR	NR	NR	NR
Schlemmer et al. (2023) [78]	12	114/123 (92.7)	73/123 (70.7)	NR	74/123 (60.2)	1/123 (0.8)	NR	71/123 (81.6)	13/123 (10.6)
Flor et al. (2023) [55]	24	18/18 (100)	1/18 (5.5)	18/18 (100)	15/18 (83.3)	0/18 (0)	NR	3/18 (16.7)	2/18 (11.1)

## Data Availability

The data presented in this study are available on request from the corresponding author.

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
