# Peer review of "Short- and Long-Term Chest-CT Findings after Recovery from COVID-19: A Systematic Review and Meta-Analysis"

_diagnostics, 2024, doi:10.3390/diagnostics14060621_

Round 1
Reviewer 1 Report
Comments and Suggestions for Authors
Summary: This systematic review and meta-analysis aims to assess the chest CT abnormalities in COVID-19 patients at different stages post-infection and by disease severity. A search was conducted for studies with follow-up chest CTs at least 12 months post-COVID-19 infection. Out of 2,517 studies, 43 were included with a total of 8,858 patients. Fibrotic-like changes were the most common in the short term and persisted in the long term. Meta-regression analysis revealed a decrease over time in the event rates for ground-glass opacities, an increase in honeycombing, but no significant change for fibrotic-like changes, bronchiectasis, reticulation, and interlobular septal thickening. Patients with severe disease had significantly higher rates of abnormalities, bronchiectasis, fibrotic-like changes, and reticulation at long-term follow-up compared to those with non-severe disease. Thus, significant CT abnormalities can persist up to 2 years post-COVID-19, particularly in patients with severe disease.
General concept comments: The study is well-conducted and written, with a clear aim and background. The introduction provides sufficient context, the materials and methods are thoroughly explained, the results are presented clearly, and the discussion is articulated well. Appropriate tables are utilized. Despite its limitations, the study effectively summarizes the common short-term and long-term changes associated with COVID-19 and underscores their significance.
Reviewer 2 Report
Comments and Suggestions for Authors
In this systematic review and meta-analysis by Babar et al., short- and long-term chest CT findings after COVID-19, stratified by disease severity, were summarized. I have the following comments:
· Did the authors categorize the patients according the inclusion criteria of the included papers or how where they able to categorize each patient?
· Did the authors adjust the results for country?
· How many of the included patients showed pre-existing pulmonary diseases? Could this be a confounder?
· Are there some information about the acute infection? How were these patients included in the studies, especially the mild ones? During hospital stays? Since the number of chest CT scans is quite high.
· Did the included studies rate the abnormalities according to severity? Since it is well known that abnormalities are also present in healthy individuals.
· Discussion: The explanation of the discrepancies in abnormalities between the current review and Watanabe et al. is not fully conclusive. A higher sample size does not automatically correspond to a higher prevalence (power calculation) and long COVID cases are usually characterized as mild acute infections, younger, and more females (what is in contrast to the population in this review) with fewer findings in clinical work ups.
· Since the prevalence of fibrotic-like changes (most prevalent abnormality) is comparable to the prevalence in influenza A, this would not support that the chest CT findings are associated with the persisting symptoms in long COVID. Or are there studies with comparable health limitations in influenza A than in COVID-19?
· Are the findings of studies with 24 months follow-up comparable to the studies with 12 months follow-up?
Round 2
Reviewer 2 Report
Comments and Suggestions for Authors
In this systematic review and meta-analysis by Babar et al., short- and long-term chest CT findings after COVID-19, stratified by disease severity, were summarized. I have the following comments to the revised version of the manuscript:
· In my opinion, it is important to discuss that the findings could not be rated according to their severity and therefore it remains open how the results can be compared to the abnormalities found in healthy ones. Or in other words, it remain to be investigated if the high prevalence of abnormal findings have a clinical relevance.
· I don’t fully understand the explanation for the comparable prevalence between SARS-CoV-2 and influenza A. The authors now disclaim a higher level of health impairments in COVID, however, shortness of breath was less common in COVID. This is contrary.
